# Learning on Graphs with Large Language Models (LLMs): A Deep Dive into Model Robustness

## Abstract

Large Language Models (LLMs) have demonstrated remarkable performance across various natural language processing tasks. Recently, several LLMs-based pipelines have been developed to enhance learning on graphs with text attributes, showcasing promising performance. However, graphs are well-known to be susceptible to adversarial attacks and it remains unclear whether LLMs exhibit robustness in learning on graphs. To address this gap, our work aims to explore the potential of LLMs in the context of adversarial attacks on graphs. Specifically, we investigate the robustness against graph structural and textual perturbations in terms of two dimensions: LLMs-as-Enhancers and LLMs-as-Predictors. Through extensive experiments, we find that, compared to shallow models, both LLMs-as-Enhancers and LLMs-as-Predictors offer superior robustness against structural and textual attacks. Based on these findings, we carried out additional analyses to investigate the underlying causes. Furthermore, we have made our benchmark library openly available to facilitate quick and fair evaluations, and to encourage ongoing innovative research in this field.

## 1 Introduction

In recent years, significant progress has been made in the development of Large Language Models (LLMs) like Sentence-BERT Reimers & Gurevych (2019), GPT Radford et al. (2018), LLaMA Touvron et al. (2023), etc. These variants showcase exceptional performance across a range of natural language processing tasks, such as sentiment analysis Sun et al. (2023b); Rønningstad et al. (2024), machine translation Feng et al. (2024); Zhang et al. (2023), and text classification Sun et al. (2023a); Zhang et al. (2024). While LLMs are widely employed for handling plain text, there is an increasing trend of applications where text data is linked with structured information represented as text-attributed graphs (TAGs) Chen et al. (2024); He et al. (2023). Recently, solely utilizing LLMs for graph data has proven effective in various downstream graph-related tasks, and integrating LLMs with Graph Neural Networks (GNNs) Kipf & Welling (2016); Veličković et al. (2017) can further enhance graph learning capabilities Chen et al. (2024).

Although graph machine learning methods with LLMs (Graph-LLMs) have reported promising performance Qin et al. (2023); Chen et al. (2023); Wei et al. (2024); Guo et al. (2023); Zhao et al. (2024); Wang et al. (2024); Cao et al. (2023); Liu et al. (2023); Qian et al. (2023); Chien et al. (2022); Duan et al. (2023), their robustness to adversarial attacks remains unknown. Robustness has always been a crucial aspect of model performance, especially in high-risk tasks like medical diagnosis Ahmedt-Aristizabal et al. (2021), autonomous driving Xiao et al. (2023), epidemic modeling Liu et al. (2024b), where failures can have severe consequences. It is well-known that graph learning models such as GNNs are vulnerable to adversarial attacks, where adversaries manipulate the graph structure to produce inaccurate predictions Zügner et al. (2018); Jin et al. (2021). Moreover, text attributes in TAGs are also vulnerable to manipulation, raising concerns about the reliability of graph learning algorithms in safety-critical applications. In the era of LLMs embracing graphs, a critical question arises: *Are Graph-LLMs robust against graph adversarial attacks?*

To address this question, we identify several research gaps in existing evaluations that hinder our understanding of current methods. These include: (1) **Limited Structural Attacks for Graph-LLMs**: Existing structural attacks are tailored for GNNs and have not been tested on Graph-LLMs, leaving it uncertain whether Graph-LLMs are sensitive to subtle structural changes within graphs.

Figure 1: An overview of our benchmark. The evaluation is divided into two perspectives: LLMs-as-Enhancers and LLMs-as-Predictors, both of which consider structural and textual attacks.

Understanding this sensitivity is crucial for developing robust Graph-LLMs. (2) **Limited Textual Attacks for TAGs**: Current feature attacks Zügner et al. (2020); Ma et al. (2020) particularly manipulate node attributes in the embedding of continuous space, rather than in textual space. This calls for creating an evaluation framework for assessing the robustness of Graph-LLMs to textual attacks. This area has been minimally explored, and it is essential to understand how text manipulations within graphs can impact model performance. (3) **Diverse Architectures of Graph-LLMs**: Various strategies exist for utilizing LLMs for graph data, with approaches such as LLMs-as-Enhancers Chen et al. (2024); He et al. (2023); Zhao et al. (2022)and LLMs-as-Predictors Chen et al. (2024); Ye et al. (2023); Chai et al. (2023) being among the most popular. This diversity necessitates the development of specialized attack pipelines to address the unique characteristics of different Graph-LLM architectures.

In response to these challenges, we aim to conduct a fair and reproducible evaluation of both structural and textual attacks under the representative pipelines of Graph-LLMs for node classification. Our contributions can be summarized as follows:

- **New evaluation perspective.** Different from existing works focusing on the predictive power of Graph-LLMs, we stress test the robustness of Graph-LLMs against graph adversarial attacks. Specifically, our work introduces a dual-focus evaluation of robustness against both structural and textual attacks, specifically targeting Graph-LLMs. Unlike traditional LLM studies Wu et al. (2024); Chao et al. (2024); Shuyuan et al.; Wang & Zhao (2024), we consider the unique challenges posed by interconnected graph structures and multi-dimensional perturbations.
- **Reproducible and comprehensive comparison.** We conduct a comprehensive comparison of various Graph-LLM pipelines across multiple datasets. Specifically, we systematically evaluate two distinct groups of Graph-LLMs: LLMs-as-Enhancers and LLMs-as-Predictors. To guarantee reproducibility and fairness in our comparison, we fine-tune all models using the same set of hyper-parameters. We employ two types of evaluation metrics: performance degradation percentage and attack success rate.
- **Insights into the robustness of Graph-LLMs:** This study reveals several interesting observations about the robustness of Graph-LLMs:
  (a) LLMs-as-Enhancers exhibit greater robustness against adversarial structural attacks compared to shallow embeddings like Bag of Words (BOW) and TF-IDF, particularly at high attack rates. Additionally, the better the distinguishability of the encoded features, the better the robustness.
  (b) LLMs-as-Predictors show better robustness in resisting structural attacks than Vanilla GNN in both zero-shot and full-shot settings.
  (c) LLMs-as-Predictors show better robustness in resisting textual attacks than MLP and the fine-tuned predictor holds even better robustness than vanilla GNN.
  (d) LLMs-as-Enhancers demonstrate excellent robustness against textual attacks, with GCN being significantly more robust than MLP as the victim model.
  (e) Text entropy, text length, and node centrality show certain negative correlations with the attack success rate of the textual attack for Graph-LLMs.
- **Open-source benchmark library:** To support and advance future research, we have developed an easy-to-use open-source benchmark library, now publicly available on https://anonymous.

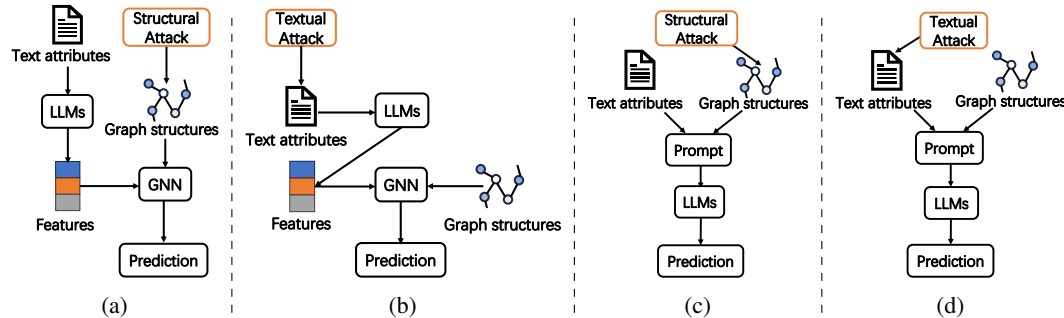

Figure 2: Evaluation pipelines: (a)(b) for LLMs-as-Enhancers in structural and textual attacks, respectively; (c)(d) for LLMs-as-Predictors in structural and textual attacks, respectively.

4open.science/r/ICLR2025 This library allows researchers to quickly evaluate their own methods or datasets with minimal effort. Additionally, we have outlined potential future directions based on our benchmark findings to inspire further investigations.

## 2 FORMULATIONS AND BACKGROUND

We begin by providing preliminaries on graph neural networks, and then formalize the graph adversarial attacks. Finally, we briefly introduce the developments in large language models on graphs. More details are shown in Appendix A.

**Notations.** We define a graph as $G = (V, E)$, where $V$ denotes the node set and $E$ represents the edge set. We employ $\mathbf{X} \in \mathbb{R}^{N \times d}$ to denote the node feature matrix, where $N$ is the number of nodes and $d$ is the dimension of the node features. Furthermore, we use the matrix $\mathbf{A} \in \mathbb{R}^{N \times N}$ to signify the adjacency matrix of $G$. Finally, the graph data can be denoted as $G = (\mathbf{A}, \mathbf{X})$.

**Graph Adversarial Attacks.** While graph adversarial attacks can perturb node features or graph structures, most existing attacks focus on modifying the graph structure due to its complexity and effectiveness. These modifications often involve adding, deleting, or rewiring edges Jin et al. (2020); Madry et al. (2017); Geisler et al. (2021); Xu et al. (2019); Chang et al. (2020); Ma et al. (2019); Entezari et al. (2020); Chen et al. (2021); Zhang et al. (2021), exemplified by the PGD Madry et al. (2017) and PRBCDGeisler et al. (2021) attacks.

**Textual Attack.** Textual attacks can be performed on different levels like character level or sentence level according to the target to be perturbed. In this work, we focus on word-level attacks, applying substitutions to fool the classifier with minimal text perturbation. For example, **SemAttack** Wang et al. (2022a) generates adversarial text by employing various semantic perturbation functions.

**Large Language Models (LLMs) on Graphs.** Recent advances in Large Language Models (LLMs) like BERT Devlin et al. (2018), Sentence-BERT (SBert) Reimers & Gurevych (2019) E5 Wang et al. (2022b), GPT Radford et al. (2018), LLaMA Touvron et al. (2023) and their variants have significantly impacted graph-related tasks. Two main paradigms are LLMs-as-Enhancers, improving node features (e.g., TAPE He et al. (2023), KEA Chen et al. (2024), GLEM Zhao et al. (2022)), and LLMs-as-Predictors, leveraging LLMs for graph predictions (e.g., InstructGLM Ye et al. (2023), GraphLLM Chai et al. (2023), GraphGPT Tang et al. (2023)).

## 3 BENCHMARK DESIGN

To deepen our understanding of the potential of Graph-LLMs in the context of robustness on graph learning, we need to design diverse pipelines to systematically assess the robustness of LLM approaches against adversarial attacks on graphs. Our benchmark evaluation encompasses two pivotal dimensions: LLMs-as-Enhancers and LLMs-as-Predictors. In this section, we will introduce the benchmark design. Details about the benchmark datasets are provided in Appendix B.

## 3.1 THREAT MODEL

We describe the characteristics of the graph adversarial attacks we developed, including both structural and textual attacks, from the following aspects. *(1) Adversary's Goal*: The primary objective is focused on evasion attacks, where the adversary seeks to manipulate the input graph data at inference time to cause the model to make incorrect predictions. In this scenario, the adversaries do not have the authority to change the classifier or its parameters. *(2) Victim Models:* The targets of these attacks include LLMs-as-Predictors and LLMs-as-Enhancers, of which the details will be thoroughly elaborated in Sections 3.2 and 3.3. *(3) Adversary's Knowledge:* The attacks are designed under white-box and grey-box frameworks, meaning that the adversary either possesses complete knowledge of the model architecture and parameters or not respectively. White-box attacks are employed during the LLM-as-Enhancers experiments to evaluate robustness in the worst-case scenarios. However, for LLM-as-Predictors, it becomes impractical to perform white-box attacks due to the significant time costs associated with these large, complex models, and thus we adopt a grey-box setting.

## 3.2 LLMS-AS-ENHANCERS

For LLMs-as-Enhancers, our benchmark provides a fair and comprehensive comparison of existing representative methods from two perspectives: structural attack and textual attack.

**Structural attack:** We have discussed two commonly used methods for structural attacks, PGD Xu et al. (2019) and PRBCD Geisler et al. (2021), in Section 2. Currently, existing frameworks rely on shallow features such as Bag of Words (BOW) and TF-IDF. In the era of LLMs, it's imperative to examine the impact of LLM features on structural attacks. Therefore, we designed a pipeline for structural attacks using LLMs-as-Enhancers. Specifically, we first generate diverse feature types derived from various LLMs, including SBert, E5, LLaMA, and Angle-LLaMA Li & Li (2023), as well as LLaMA fine-tuned (LLaMA-FT) with LoRa Hu et al. (2021), etc. We then utilize these features to evaluate the performance of structural attacks. The pipeline is visualized in Figure 2(a).

**Textual attack:** We conduct an evaluation on text attacks to verify whether LLMs-as-Enhancers can withstand textual attacks compared to traditional text preprocessing techniques. Specifically, in the white-box setting, we first conduct text attacks by using **SemAttack** Wang et al. (2022a) on the texts of text-attributed graphs. Then, we encode the texts using different methods such as traditional techniques and LLMs. Finally, we assess their performance on GCN Kipf & Welling (2016) and MLP. Incidentally, to enhance efficiency on LLMs, we modified SemAttack for batch-wise operation instead of word-level processing. The pipeline is illustrated in Figure 2(b).

## 3.3 LLMS-AS-PREDICTORS

For LLMs-as-Predictors, we also perform structural and textual attacks on pre-trained and fine-tuned LLMs, respectively. Different from the attacks on LLMs-as-Enhancers, white-box attacks on the predictors can bring enormous computational costs due to the complexity of the models. Therefore, this study adopts the grey-box setting, choosing LLMs-as-Enhancers as the victim model and transferring the attacked texts or graphs to the LLMs-as-Predictors. For pre-trained models, we utilize the same pipeline in Chen et al. (2024), which describes the graph structure in text and inputs it along with text features directly into GPT-3.5 for prediction. For fine-tuned models, we perform attacks on InstructGLM Ye et al. (2023), which uses LLaMA Touvron et al. (2023) as the backbone and is fine-tuned on different benchmark datasets.

**Structural attack:** Although the LLMs-as-Predictors in this study flatten graph structure into texts and only incorporate a small number of neighbors when predicting a node, it is possible that introducing irrelevant or false neighbors can influence the prediction results. During the structure attack, we use PRBCD as the attack algorithm and choose SBert and GCN as the surrogate model. The perturbed graph then serves as the input of GPT-3.5 or InstructGLM. The pipeline is depicted in Figure 2(c).

**Textual attack:** The LLMs-as-Predictors directly utilize texts as inputs. Thus, perturbing the input texts is also likely to have impacts to the prediction results. In this study, we first perform SemAttack on the selected nodes with SBert and GCN as the surrogate model. After that, the perturbed texts are used to evaluate GPT-3.5 and InstructGLM. The pipeline is shown in Figure 2(d).

Table 1: Performance of LLMs-as-Enhancers against **0%, 5% and 25% structural attacks**. **GAP** represents the percentage (%) decrease in performance after an attack compared to the clean performance. N/A indicates that TAPE does not provide the explanation features. We use pink to denote the best performance, green for the second-best, and yellow for the third-best.

| Dataset | Ptb. | BOW ACC | GAP | TF-IDF ACC | GAP | SBert ACC | GAP | E5 ACC | GAP | LLaMA ACC | GAP | Angle-LLaMA ACC | GAP | Explanation ACC | GAP | Ensemble ACC | GAP | LLaMA-FT ACC | GAP |
|---|---|---|---|---|---|---|---|---|---|---|---|---|---|---|---|---|---|---|---|
| Pudmed | 0% | 74.69 | 0% | 76.86 | 0% | 78.71 | 0% | 81.83 | 0% | 77.65 | 0% | 75.15 | 0% | 88.84 | 0% | 83.59 | 0% | 77.4 | 0% |
|  | 5% | 72.40 | 3.07% | 74.68 | 2.8% | 74.68 | 5.1% | 79.39 | 3.0% | 76.68 | 1.3% | 74.65 | 0.67% | 87.32 | 1.5% | 82.55 | 1.2% | 76.55 | 1.1% |
|  | 25% | 62.83 | 24.30% | 66.18 | 13.9% | 69.24 | 12.0% | 71.06 | 13.1% | 74.62 | 3.9% | 72.72 | 3.2% | 82.11 | 7.6% | 77.85 | 6.9% | 75.63 | 2.3% |
| Arxiv | 0% | 50.99 | 0% | 48.39 | 0% | 52.51 | 0% | 57.04 | 0% | 58.04 | 0% | 58.53 | 0% | 54.37 | 0% | 59.9 | 0% | 52.46 | 0% |
|  | 5% | 42.35 | 16.9% | 43.32 | 10.5% | 47.59 | 9.4% | 48.15 | 15.6% | 51.76 | 10.8% | 51.46 | 12.1% | 47.19 | 13.2% | 54.33 | 9.3% | 47.59 | 9.3% |
|  | 25% | 18.95 | 62.9% | 24.36 | 49.7% | 29.24 | 44.3% | 25.24 | 55.8% | 31.67 | 45.4% | 31.73 | 45.8% | 21.61 | 60.3% | 32.82 | 45.2% | 29.19 | 44.4% |
| Cora | 0% | 78.49 | 0% | 81.46 | 0% | 81.99 | 0% | 83.17 | 0% | 78.13 | 0% | 80.28 | 0% | 82.79 | 0% | 82.57 | 0% | 78.54 | 0% |
|  | 5% | 73.71 | 3.1% | 77.23 | 5.2% | 79.18 | 3.4% | 80.43 | 3.3% | 70.91 | 9.2% | 80.23 | 0.06% | 80.47 | 2.8% | 80.67 | 2.3% | 78.51 | 0% |
|  | 25% | 60.72 | 22.6% | 67.53 | 17.1% | 72.14 | 12.0% | 69.84 | 16.0% | 69.99 | 10.4% | 76.75 | 4.4% | 71.32 | 13.9% | 74.89 | 9.3% | 70.72 | 10.0% |
| WikiCS | 0% | 74.92 | 0% | 75.96 | 0% | 75.89 | 0% | 76.71 | 0% | 79.72 | 0% | 73.33 | 0% | N/A | N/A | N/A | N/A | 79.69 | 0% |
|  | 5% | 61.46 | 18.0% | 62.03 | 18.3% | 64.44 | 15.1% | 63.01 | 17.9% | 70.91 | 11.1% | 67.0 | 8.6% | N/A | N/A | N/A | N/A | 70.11 | 12.0% |
|  | 25% | 45.34 | 39.5% | 45.74 | 39.8% | 50.19 | 33.9% | 46.38 | 39.6% | 59.23 | 25.7% | 56.75 | 22.6% | N/A | N/A | N/A | N/A | 60.13 | 24.6% |
| History | 0% | 51.25 | 0% | 58.38 | 0% | 65.59 | 0% | 66.96 | 0% | 66.73 | 0% | 66.52 | 0% | N/A | N/A | N/A | N/A | 64.53 | 0% |
|  | 5% | 49.04 | 4.3% | 57.57 | 1.4% | 65.6 | 0% | 64.69 | 3.4% | 64.69 | 7.6% | 63.47 | 4.6% | N/A | N/A | N/A | N/A | 62.99 | 2.4% |
|  | 25% | 38.93 | 24.0% | 22.66 | 61.2% | 56.34 | 14.1% | 55.37 | 17.3% | 54.54 | 18.3% | 52.75 | 20.7% | N/A | N/A | N/A | N/A | 53.34 | 17.4% |
| Citeseer | 0% | 70.74 | 0% | 73.02 | 0% | 74.94 | 0% | 75.14 | 0% | 67.48 | 0% | 71.71 | 0% | N/A | N/A | N/A | N/A | 69.7 | 0% |
|  | 5% | 68.84 | 2.7% | 71.40 | 2.2% | 72.60 | 3.1% | 72.29 | 3.8% | 70.82 | 5.0% | 68.46 | 4.5% | N/A | N/A | N/A | N/A | 68.62 | 1.6% |
|  | 25% | 61.76 | 12.7% | 64.36 | 11.9% | 64.51 | 13.9% | 64.38 | 14.3% | 64.69 | 4.1% | 68.61 | 4.3% | N/A | N/A | N/A | N/A | 66.17 | 5.1% |

## 4 EXPERIMENTS

In this section, we assess the robustness of LLMs against graph adversarial attacks in their two roles: LLMs-as-Enhancers and LLMs-as-Predictors. Specifically, we aim to answer the following questions: **Q1:** How effective are the LLMs-as-Enhancers on structural attack? **Q2:** What is the effectiveness of LLMs-as-Enhancers on textual attacks? **Q3:** How effective are the LLMs-as-Predictors on structural attack? **Q4:** How do LLMs-as-Predictors perform on textual attacks? The experimental settings and usage instructions are reported in Appendix C.

### 4.1 STRUCTURAL ATTACK FOR LLMS-AS-ENHANCERS

**Experiment Design.** To tackle the research question **Q1**, we enhance text attributes using LLMs and generate new features. These enriched features are then used to train a GCN as the victim model. Specifically, we use PGD to conduct white-box evasion attacks on the structures of small graphs such as Cora, Citeseer, Pubmed, and Wikics, while we employ PRBCD to conduct white-box evasion attacks on the structures of large graphs Arxiv and History. We vary the perturbation rates at 0% (clean graphs), 5%, and 25%, which represent the ratio of perturbed edges to original edges. Subsequently, we feed the perturbed graphs into different LLMs-as-Enhancers architectures and compare the LLM features with shallow features. To quantify model robustness, we report the test accuracy (ACC) and the percentage accuracy degradation after attacks compared to the original accuracy (GAP).

**Results.** The results are reported in Table 1. Further details, including standard deviations, can be found in Appendix D. In addition, we incorporate additional GNN architectures, such as GraphSAGE, in our experiments to evaluate the robustness against adversarial attacks. The time costs are provided in Appendix L. The results of Deepseek Liu et al. (2024a) are shown in Table 20. Lastly, we conducted PGD and PRBCD attacks on the Cora, Citeseer and Pubmed and found that the performance of large language models remained consistent against both attacks. The detailed data and conclusions are provided in Appendix J. From these results, we have the following observations.

*Performance comparison on clean graphs.* The features generated by pre-trained language models exhibit better performance on most clean datasets. For instance, SBert and e5-large (E5) show improvements of 2.4% and 6.5% respectively on clean Pubmed datasets compared to TF-IDF.

*Robustness against structural attacks.* For 25% evasion attacks, almost all language models exhibit greater robustness compared to traditional BOW and TF-IDF approaches. For example, in the case of a 25% evasion attack on Pubmed, while TF-IDF experiences a decrease of 13.9%, LLaMA only drops by 3.9%. Moreover, we observe that fine-tuning enhances the robustness of LLMs, as demonstrated by the fact that fine-tuned LLaMA exhibits greater robustness compared to its unfine-tuned counterpart. Further, by comparing the results on 5% and 25% attacks, we find that LLMs-as-Enhancers are more

Table 2: Performance (ACC) for various models on different datasets and structural perturbation levels.

| GraphSAGE | Ptb. | BOW | TFIDF | SBert | E5 | LLaMA | Angle-LLaMA |
|---|---|---|---|---|---|---|---|
| Cora | 0 | 79.11 ± 1.05 | 80.34 ± 1.32 | 80.92 ± 0.89 | **81.87 ± 1.13** | 79.36 ± 1.07 | 80.54 ± 0.31 |
| | 5% | 74.55 ± 1.29 | 77.40 ± 1.36 | 77.89 ± 0.49 | **79.56 ± 1.40** | 75.99 ± 0.32 | 77.09 ± 0.88 |
| | 25% | 65.40 ± 2.67 | 66.95 ± 2.40 | 71.73 ± 0.77 | **73.74 ± 2.02** | 71.34 ± 0.95 | 73.04 ± 1.26 |
| Citeseer2 | 0 | 69.43 ± 1.32 | 71.75 ± 1.70 | **75.25 ± 1.24** | 73.61 ± 1.27 | 71.84 ± 1.21 | 73.10 ± 0.96 |
| | 5% | 68.22 ± 1.73 | 70.72 ± 1.35 | **73.40 ± 1.75** | 72.99 ± 0.83 | 70.93 ± 1.54 | 71.83 ± 0.57 |
| | 25% | 63.09 ± 2.17 | 66.42 ± 1.59 | **70.48 ± 1.80** | 68.36 ± 1.68 | 68.85 ± 1.67 | 69.15 ± 2.48 |
| Pubmed | 0 | 72.78 ± 2.13 | 74.60 ± 1.07 | 77.68 ± 0.54 | **78.95 ± 1.20** | 74.68 ± 1.09 | 69.89 ± 2.33 |
| | 5% | 69.10 ± 1.30 | 70.95 ± 1.23 | 74.69 ± 0.77 | **76.32 ± 1.74** | 73.15 ± 0.40 | 66.40 ± 1.86 |
| | 25% | 62.79 ± 2.46 | 64.63 ± 1.59 | 70.18 ± 1.47 | **71.91 ± 1.59** | 69.30 ± 2.38 | 64.15 ± 2.87 |

helpful at higher perturbation rates. For example, with a 5% perturbation budget on Cora, only 4 language models show a lower GAP compared to the shallow BOW features, while this number increases to 7 at the perturbation rate of 25%. However, by examining the accuracy on clean graphs, we note that the performance of LLM features still declines considerably on certain datasets, such as Arxiv, under high perturbation rates, indicating that graph LLMs remain vulnerable to attacks. Furthermore, we incorporate additional dataset, such as Arxiv23, into our experiments, where certain portions of this dataset have not been previously encountered by LLMs. The results and conclusion are presented in Appendix M.

*Results for GraphSAGE.* We incorporate additional GNN architectures, such as GraphSAGE, in our experiments to evaluate the robustness against adversarial attacks. From the results in Table 2, we find that when GraphSAGE is used for prediction, the features generated by the large language model can still help the GNN achieve better adversarial robustness, which is consistent with previous conclusions on GCN.

*Robustness of explanation features.* "Explanation" refers to the explanation features generated by the TAPE model He et al. (2023). Specifically, the original text features are processed by TAPE, from which the new generated texts are served as the augmented inputs. Similarly, "Ensemble" denotes the combined use of both explanation features and LLaMA features of the original inputs. As shown in the Table 1, we find that the use of these augmented features improves the model's robustness.

**Key Takeaways 1:** Most LLMs demonstrate greater robustness against structural attacks compared to shallow models. The analysis can be found in Section 4.5.

**Key Takeaways 2:** The higher the attack rate, the more robust the features of LLMs compared to shallow features.

## 4.2 TEXTUAL ATTACK FOR LLMS-AS-ENHANCERS

**Experiment Design.** To answer **Q2**, we conduct white-box evasion attacks on LLMs-as-Enhancers, targeting textual attributes. We first utilize diverse LLMs-as-Enhancers to transform the text into node embeddings and train a GCN and an MLP. Then we perform the evasion attack at the model inference stage by perturbing text using SemAttack. Then, LLMs are used to transform the perturbed text into node embeddings, which will then be fed into the trained GCN or MLP for inference. For all models and datasets, we randomly sample 200 nodes as target nodes and utilize the Attack Success Rate (ASR) Wang et al. (2022a) as the evaluation metric.

**Results.** The results are reported in Table 3. Additional details, including standard deviations of performance, are provided in Appendix E. From these results, we can draw the following observations.

*LLaMA performs well against textual attack when MLP is used as the victim model.* When using MLP as the victim model, E5 and LLaMA demonstrate greater resilience against SemAttack, with a noticeable downward trend in ASR for SBert, E5, and LLaMA models. For example, on the WikiCS dataset, SBert has an ASR of 62.45%, while the performance of LLaMA dropped to 22.17%. Another interesting observation is that BOW shows better robustness than SBert on Cora, Pubmed, and Arxiv. Given that BOW has a limited input of words, the robustness of BOW is likely to come from filtering the perturbed words, whose frequencies are often low.

Table 3: Performance of LLMs-as-Enhancers against the textual attack. Bold numbers represent the lowest Attack Success Rate (ASR), indicating superior robustness.

| Features | Cora | | Pubmed | | Arxiv | | Wikics | | History | | Citeseer | |
|---|---|---|---|---|---|---|---|---|---|---|---|---|
| | MLP | GCN | MLP | GCN | MLP | GCN | MLP | GCN | MLP | GCN | MLP | GCN |
| BOW | 62.11 | 9.27 | 41.00 | 8.58 | 72.69 | 15.19 | 67.85 | 3.82 | 74.53 | 18.40 | 68.80 | 18.98 |
| SBert | 73.18 | 14.76 | 45.36 | 9.32 | 82.69 | 11.17 | 62.12 | 9.08 | 76.73 | 13.41 | 66.33 | 16.93 |
| E5 | 65.29 | 10.53 | 35.62 | 8.76 | 81.92 | 15.22 | 65.99 | 6.64 | 61.51 | **6.54** | 57.10 | 14.69 |
| LLaMA | 56.49 | 12.58 | 19.66 | 4.95 | 67.87 | **6.05** | **22.17** | 4.17 | 65.07 | 15.92 | 46.81 | 13.77 |
| LLaMA-FT | **40.10** | **4.37** | **16.69** | **3.27** | **67.71** | 6.08 | 30.00 | **2.98** | **56.98** | 6.91 | **38.46** | **6.58** |

Table 4: The robustness of the predictor InstructGLM against **5% structural attack**. **GAP** refers to the percentage (%) performance decrease after an attack compared to the clean ACC. The bold font is used to highlight the smallest gap.

| Dataset | Cora | | | Pubmed | | | Arxiv | | |
|---|---|---|---|---|---|---|---|---|---|
| model \ Perturbation Rate | Clean | Attack | GAP | Clean | Attack | GAP | Clean | Attack | GAP |
| GCN | 87.45 | 81.73 | 6.54% | 87.07 | 84.25 | 3.24% | 57.5 | 52.94 | 7.93% |
| InstructGLM (structure-aware) | 82.47 | 80.73 | **2.11%** | 91.63 | 91.10 | **0.58%** | 72.87 | 71.84 | **1.41%** |

*For textual attack, GCN as the victim model is more robust compared to MLP as the victim model.* In terms of GCN as the victim model, fine-tuned LLaMA achieves the lowest ASR among all datasets, ranging from 2.98% to 6.91%. Also, compared to MLP, the ASR for GCN decreases significantly and remains below 20% across all datasets.

**Key Takeaways 3:** Among all models and settings, the fine-tuned model, LLaMA-FT, generally exhibits the best robustness against the textual attack on most datasets. The detailed analysis is provided in Section 4.6.

**Key Takeaways 4:** In the LLMs-as-Enhancers framework, GCN greatly improves the robustness against textual attacks compared to MLP as the victim model. The analysis is presented in Section 4.6.

### 4.3 STRUCTURAL ATTACK FOR LLMS-AS-PREDICTORS

**Experiment Design.** To answer **Q3**, we explore the robustness of LLM-as-Predictors against graph structural attacks, using GPT-3.5 and InstructGLM as the selected predictors. Given the difficulty in directly attacking these predictors due to their large number of parameters or lack of model access, we adopt a grey-box setting. In this setting, LLMs-as-Enhancers are used as surrogate models during adversarial attacks, and the resulting attacked graph structures are then fed into the LLMs-as-Predictors. When employing GPT-3.5 in the graph domain, we follow the approach in Chen et al. (2024) and evaluate 200 nodes in the test set. For InstructGLM, we directly attack the pre-processed datasets provided by the author.

**Results.** The results of InstructGLM are reported in Table 4. Additional results for GPT-3.5 are presented in Figure 3, while further results for GPT-4o Mini can be found in Table 15 in Appendix H. Based on these results, we can make the following observations.

*GPT-3.5 shows the strongest robustness against structural attack in the zero-shot setting.* We evaluate the model under various few-shot and zero-shot settings with the input of summarized two-hop neighbors, as illustrated in Chen et al. (2024). The results demonstrate that GPT-3.5 maintains the highest robustness in the zero-shot setting, with minimal performance degradation even under significant perturbations. We conjecture that the neighbor sampling and summarizing processes likely mitigate noise introduced by structural alterations. Although GAT also uses the migrated attacked graph structure from GCN, its accuracy drops much faster than that of GPT-3.5.

*Similarly, InstructGLM also shows stronger robustness against structural attack compared to GCN.* While GCN experiences a noticeable accuracy decrease after 5% structural perturbation, InstructGLM maintains performance close to that on the clean graph. Like GPT-3.5, InstructGLM also employs neighbor sampling and its robustness may benefit from this procedure.

**Key Takeaways 5:** LLMs-as-Predictors show strong robustness against the structural attack, especially in the zero-shot setting. The analysis is shown in Appendix I.

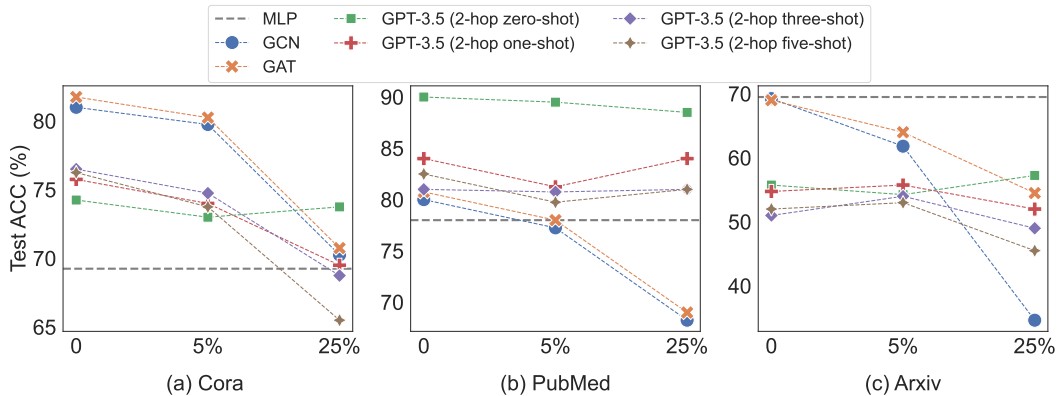

Figure 3: The performance of LLMs-as-Predictors against structural attacks, evaluated by accuracy.

Table 5: The performance of the predictor GPT-3.5 against **5%** textual attacks. The bold font is used to highlight the lowest Attack Success Rate.

| Dataset | Cora | | Pubmed | | Arxiv | |
|---|---|---|---|---|---|---|
| Model | Clean ACC | ASR | Clean ACC | ASR | Clean ACC | ASR |
| MLP | 69.25 | 35.97 | 78.00 | 14.74 | 69.50 | 30.10 |
| GCN | 81.00 | 4.30 | 80.00 | 3.20 | 69.25 | 5.42 |
| GAT | 81.75 | **2.99** | 80.75 | **2.23** | 69.00 | **3.97** |
| GPT-3.5 (2-hop zero-shot) | 74.25 | 10.03 | 90.00 | 5.53 | 55.75 | 13.33 |
| GPT-3.5 (2-hop one-shot) | 75.75 | 5.59 | 84.00 | 3.75 | 54.75 | 18.65 |
| GPT-3.5 (2-hop three-shot) | 76.50 | 5.59 | 81.00 | 5.29 | 51.00 | 19.10 |
| GPT-3.5 (2-hop five-shot) | 76.25 | 6.72 | 82.50 | 6.07 | 52.00 | 21.24 |

## 4.4 TEXTUAL ATTACK FOR LLMS-AS-PREDICTORS

**Experiment Design.** To answer **Q4**, we explore the performance of GPT-3.5 and InstructGLM with perturbed texts as inputs. The perturbed texts are generated by SemAttack and used as adversarial inputs for GPT-3.5 and InstructGLM to evaluate the robustness of LLMs-as-Predictors against textual attacks. Following the experiment design in **Q3**, the experiment is conducted in a grey-box setting. this experiment is conducted in a grey-box setting. We use GCN with SBert embeddings as the surrogate model, randomly sampling 200 target nodes for attack and evaluation.

**Results.** The results about GPT-3.5 are presented in Table 5, More results of InstructGLM are presented in Table 6, while further results for GPT-4o Mini and LLaGA under SemAttack attack Wang et al. (2022a) can be found in Table 16 in Appendix H. For the **sentence-level attack SCPN** Iyyer et al. (2018) on GPT-4o Mini, results are available in Appendix N. These results lead us to the following observations.

*GPT-3.5 shows stronger robustness compared to MLP but failed to exceed GCN.* As shown in Table 5, our experiment reveals that GPT-3.5 has strong robustness compared to MLP but fails to exceed GCN and GAT in all few shot settings. On the other hand, GAT exhibits the strongest robustness, followed by GCN. However, on Cora and Pubmed, the Attack Success rate of GPT-3.5 is close to GCN, ranging from 0.55% to 1.70%.

*InstructGLM shows the strongest robustness compared to both MLP and GCN.* Different from the above observations of GPT-3.5, the fine-tuned InstructGLM shows stronger robustness on the three datasets compared to GCN. While there is an ASR of 30.37% for GCN on Cora, InstructGLM can resist the textual perturbation and achieves an ASR of 1.21% with structural information incorporated. The 0% ASR result of MLP on the Arxiv dataset is due to the low ACC, as we can not find a sample that is both predicted correctly before the attack and predicted wrong after the attack. Also, this low accuracy is likely to be caused by the dataset used by InstructGLM, which only utilizes titles of a few words as the node attributes.

Table 6: The robustness of the predictor InstructGLM against textual attacks. The bold font is used to highlight the lowest Attack Success Rate (ASR).

| Dataset | Cora | | Pubmed | | Arxiv | |
|---|---|---|---|---|---|---|
| Model | Clean ACC | ASR | Clean ACC | ASR | Clean ACC | ASR |
| MLP | 67.53 | 30.37 | 86.92 | 10.11 | 5.86 | **0** |
| GCN | 88.01 | 10.22 | 86.82 | 3.55 | 58.85 | 6.61 |
| InstructGLM (structure-aware) | 82.50 | **1.21** | 91.50 | **1.09** | 74.87 | 3.42 |

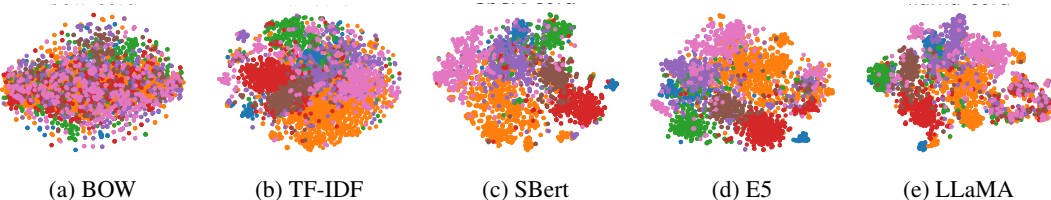

| (a) BOW | (b) TF-IDF | (c) SBert | (d) E5 | (e) LLaMA |
|---|---|---|---|---|

Figure 4: t-SNE visualization of initial Cora features, with different colors representing classes.

**Key Takeaways 6:** LLMs-as-Predictors are more robust against textual attacks than MLP. However, GPT-3.5, which is not fine-tuned, shows poorer robustness compared to GCN. Similarly, the analysis in Section 4.6 provides support for explaining why GCN is more robust than MLP.

### 4.5 ANALYSIS FOR STRUCTURAL ATTACK

Since Sections 4.1 and 4.3 have shown the robustness of Graph-LLMs against the structure attack, we conduct analysis from the following perspectives to explore the reasons behind such robustness.

**t-SNE visualization.** First, we examine t-SNE visualizations of various initial features and find that initial features generated by language models are more distinguishable in categories compared to traditional features as shown in Fig. 4.

**DBI.** To further evaluate the separability of input features, we use the Davies-Bouldin Index (DBI) Davies & Bouldin (1979), as shown in Table 7. The DBI score represents the average similarity measure between each cluster and its most similar cluster, with lower scores indicating better clustering quality. A score of zero is ideal, signifying optimal clustering. We find that initial LLM features have lower initial DBI scores. We further examine the DBI of embeddings both before and after the attack, and the differences therein. Notably, embeddings from larger models exhibit a smaller decrease in DBI scores post-attack, as indicated by the DBI Diff in the Table 7.

**Homophily.** Additionally, we analyze the homophily of the Cora dataset before and after attacks, as shown in Table 7. It discovers that the pre-attack Cora dataset has a homophily of 0.81, and after the attack, Cora with shallow features exhibits lower homophily.

Based on the above results, we find that robustness is strongly positively correlated with the quality of features, indicating that higher distinguishability of features leads to stronger robustness and higher homophily after attacks. This could be attributed to the richer information present in features generated by pre-trained language models, resulting in higher distinguishability in clustering. The higher the quality of the features, the less the model depends on the structure. Therefore, high-quality features of LLMs can be more robust against structural attacks.

### 4.6 ANALYSIS FOR TEXTUAL ATTACK

From the experiments of the LLMs-as-Enhancers against textual attack in Section 4.2, we observe that LLaMA performs much better than other smaller models and the **fine-tuned LLaMA** exhibits the best robustness. In addition, GCN demonstrates much stronger robustness compared to MLP. To explore the reasons behind this, we perform analysis from feature and structure perspectives.

**Feature Perspective.** From the feature perspective, we assume that text attributes can have an impact on the attack success rate. Specifically, we first explore some basic indicators like text entropy, text length, and the number of words among successfully attacked nodes and failed attacked nodes, as shown in Table 8. Based on the observation with LLaMA and GCN as the victim model, it is obvious

Table 7: DBI and homophily of Cora. "Init DBI" indicates the DBI of initial features, "Embed DBI" represents the DBI of trained embedding, "Post-Attack DBI" refers to the DBI of embedding after an attack, "DBI Diff" donates the difference between Embed DBI and Post-Attack DBI. The homophily value of the graph before being attacked is 0.81.

| | GCN | MLP | Init DBI↓ | Embed DBI↓ | Post-Attack DBI↓ | DBI Diff↓ | GAP↓ | Homophily (0.81) |
|---|---|---|---|---|---|---|---|---|
| BOW | 78.49 | 55.90 | 9.34 | 1.60 | 2.75 | 1.15 | 22.68% | 0.66 |
| TF-IDF | 81.46 | 65.85 | 8.85 | 1.24 | 1.89 | 0.65 | 17.10% | 0.66 |
| SBert | 81.99 | 70.80 | 4.47 | 1.28 | 1.56 | 0.28 | 12.00% | 0.70 |
| E5 | 83.17 | 69.16 | 5.92 | 1.27 | 1.66 | 0.39 | 16.00% | 0.67 |
| LLaMA | 78.13 | 67.48 | 4.88 | 1.83 | 1.90 | 0.07 | 10.41% | 0.68 |

Table 8: Comparisons between successfully attacked nodes and failed attacked nodes from the feature perspective (**success / failed**), with LLaMA and GCN as the victim model.

| Dataset | Pubmed | | Citeseer | | History | |
|---|---|---|---|---|---|---|
| | Fine-tuned | w/o Fine-tuned | Fine-tuned | w/o Fine-tuned | Fine-tuned | w/o Fine-tuned |
| Entropy | 6.25/6.40 | 6.24/6.41 | 5.90/6.09 | 5.86/6.09 | 5.45/6.21 | 5.28/6.25 |
| Text Length | 206.10/237.22 | 197.94/237.64 | 125.17/148.94 | 124.07/149.38 | 114.71/250.45 | 95.51/237.71 |
| Words | 108.48/122.04 | 108.69/122.89 | 78.60/90.80 | 77.57/90.99 | 73.45/139.59 | 63.79/138.39 |
| DBI | 4.77 | 4.87 | 5.01 | 5.24 | 4.65 | 5.02 |

that the successfully attacked nodes tend to have smaller entropy (less richness of texts), shorter texts, and smaller amounts of words. By comparing the DBI of the fine-tuned and not fine-tuned model, we also observe that the fine-tuned model always has a smaller DBI compared to the fine-tuned model.

**Structure Perspective.** From the structure perspective, we investigate the relations between the degree centrality of nodes and the attack success rate. As shown in Fig. 5, we use SBert and GCN as the victim model and visualize the distribution of degrees from successfully attacked nodes and failed attacked nodes respectively. The results clearly show that successfully attacked nodes often have smaller degrees, indicating that **nodes with less structural information in the graph are more vulnerable to textual attacks**. Additionally, similar patterns are observed for eigenvector centrality and PageRank values, which we detail in Appendix F.

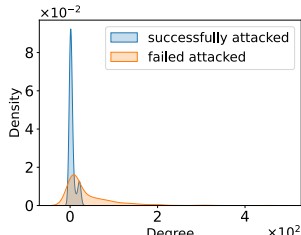

Figure 5: Centrality distributions of nodes being attacked successfully and unsuccessfully on WikiCS.

## 5 CONCLUSION AND FUTURE DIRECTIONS

This work introduces a comprehensive benchmark for exploring the potential of LLMs in context of adversarial attacks on graphs. Specifically, we investigate the robustness against graph structural and textual attacks in two dimensions: LLMs-as-Enhancers and LLMs-as-Predictors. Through extensive experiments, we find that, compared to shallow models, both LLMs-as-Enhancers and LLMs-as-Predictors offer superior robustness against structural and textual attacks. Despite these promising results, several critical challenges and research directions remain worthy of future investigation.

**Rethinking Textual Attack.** Based on the observations above, we realize that textual attacks can significantly affect the prediction of individual samples. However, when GCN serves as the victim model, the incorporation of neighbor information helps mitigate these perturbations, significantly reducing the attack's effectiveness. From an attack perspective, weakening the resistance of GCN-based victim models is crucial, particularly when targeting stronger Graph-LLMs.

**Combining Textual and Structural Attack on Graphs.** To enhance attack capabilities, a combined framework that perturbs both text attributes and graph structure is needed. However, challenges such as integrating textual and structural attacks to improve attack efficiency remain unsolved. In this study, we provide preliminary results in the Appendix G. Our experiment shows that adding additional textual perturbations on top of structural perturbations can further degrade model performance.

**Rethinking Graph-LLMs.** From the results in Table 1, we can conclude that the angle-optimized Angle-LLaMA, which is more suitable for text encoding, exhibits better robustness against adversarial attacks compared to LLaMA. This phenomenon may inspire us to design better Graph-LLMs for text attribute encoding. Finally, we can use LLMs to perform attacks on graphs by generating harmful structures and text attributes.

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

# A   FORMULATIONS AND BACKGROUND

We begin by providing preliminaries on graph neural networks, and then formalize the graph adversarial attacks. Finally, we introduce the developments in large language models on graphs.

**Notations.** We define a graph as $G = (V, E)$, where $V$ denotes the node set and $E$ represents the edge set. We employ $\mathbf{X} \in \mathbb{R}^{N \times d}$ to denote the node feature matrix, where $N$ is the number of nodes and $d$ is the dimension of the node features. Furthermore, we use the matrix $\mathbf{A} \in \mathbb{R}^{N \times N}$ to signify the adjacency matrix of $G$. Finally, the graph data can be denoted as $G = (\mathbf{A}, \mathbf{X})$.

**Graph Neural Networks.** GCN Kipf & Welling (2016) is one of the most representative models of GNNs, utilizing aggregation and transformation operations to model graph data. Unlike GCN, which treats all neighbors equally, GAT Veličković et al. (2017) assigns different weights to different nodes within a neighborhood during aggregation.

**Graph Adversarial Attacks.** In the context of $G = (\mathbf{A}, \mathbf{X})$ and a subset $V_m \subseteq V$ comprising victim nodes, where $y_i$ signifies the label for node $i$, the attacker's objective is to discern a perturbed graph denoted as $\tilde{G} = (\tilde{\mathbf{A}}, \tilde{\mathbf{X}})$. The primary goal is to minimize the attack objective $\mathcal{L}_{attack}$.

$$\min \mathcal{L}_{\text{attack}} \left( f_\theta(\tilde{G}) \right) = \sum_{i \in V_m} \ell_{\text{attack}} \left( f_{\theta^*}(\tilde{G})_i, y_i \right) \text{ s.t., } \theta^* = \arg\min_\theta \mathcal{L}_{\text{train}} \left( f_\theta \left( \hat{G} \right) \right), \quad (1)$$

where $f_\theta$ indicates the model function of GNN, $\mathcal{L}_{attack}$ represents the loss function for attacks, one option is to set $\mathcal{L}_{attack} = -\mathcal{L}$, and $\hat{G}$ can be either $G$ or $\tilde{G}$. Here, $\tilde{G}$ is chosen from a constrained domain $\Psi(G)$. Given a fixed perturbation budget $\mathbb{D}$, a typical constraint for $\Psi(G)$ can be expressed as $\|\tilde{\mathbf{A}} - \mathbf{A}\|_0 + \|\tilde{\mathbf{X}} - \mathbf{X}\|_0 \leq \mathbb{D}$. This constraint implies that the perturbations introduced in the adjacency matrix $\tilde{\mathbf{A}}$ and node feature matrix $\tilde{\mathbf{X}}$ should be limited, and their combined $L_0$ should not exceed the specified budget $\mathbb{D}$.

While graph adversarial attacks can perturb either node features or graph structures, the complexity of structural information has led the majority of existing adversarial attacks on graph data to focus on modifying graph structure, particularly through actions such as adding, deleting, or rewiring edges Jin et al. (2020); Madry et al. (2017); Geisler et al. (2021); Xu et al. (2019); Chang et al. (2020); Ma et al. (2019); Entezari et al. (2020); Chen et al. (2021); Zhang et al. (2021). For example, the **PGD** attack Madry et al. (2017) uses edge perturbation to overcome the challenge of attacking discrete graph structures via first-order optimization. In contrast, the **PRBCD** attack Geisler et al. (2021) addresses the high cost of adversarial attacks on large graphs with a sparsity-aware optimization approach.

On one hand, we explore the robustness against structural attacks. On the other hand, instead of targeting continuous features as in existing feature attack works, we adopt a direct approach by employing textual attacks to evaluate robustness, which remains a relatively unexplored direction.

**Textual Attack.** For tasks on TAGs, the raw inputs are in text format and it can be hard for attackers to manipulate the encoded features directly, which makes the traditional feature attacking on graphs less practical. Therefore, the textual attack is used in this study to evaluate the robustness of LLMs enhanced graph features. Textual attacks can be performed on different levels like character level or sentence level according to the target to be perturbed. In this work, we focus on word-level attacks, which can be defined as follows.

*Word-level attacks.* Given a classifier $f$ that predicts labels $y \in Y$, the input $X$ is defined in a categorical space and each input is a sequence of $n$ words $x_1, x_2, \cdots, x_n$. Each word $x_i$ has a limited amount of substitution candidates, denoted as $S(x)$. To keep the perturbation as unnoticeable as possible, a constraint, usually $L_1$ or $L_2$ distance is applied. Finally, to fool classifier $f$ to the largest extent, the following objective is maximized:

$$\arg\max_{x' \in S(x)} L(f(x'), y) \\ \text{s.t., } \|x' - x\|_2 < \epsilon, \quad (2)$$

where $L$ is the loss function between original prediction $y$ and prediction $f(x')$ after perturbation. $\epsilon$ is the budget of the perturbation, which keeps the original and the perturbed sample as close as possible. As the loss function above is maximized, we will get a perturbed sample $x'$ that makes the classifier $f$ generate a prediction far from the original output. For example, **SemAttack** Wang et al. (2022a) generates natural adversarial text by employing various semantic perturbation functions.

**Large Language Models on Graphs.** In recent years, remarkable progress has been achieved in the field of Large Language Models (LLMs), with notable contributions from transformative architectures such as Transformers Reimers & Gurevych (2019), BERT Devlin et al. (2018), Sentence-BERT (SBert) Reimers & Gurevych (2019) E5 Wang et al. (2022b), GPT Radford et al. (2018), LLaMA Touvron et al. (2023) and their variants. These LLMs can be applied to graph-related tasks. The collaboration between Large Language Models and Graph Neural Networks (GNNs) can be mutually advantageous, leading to improved graph learning. The two most popular paradigms for applying LLMs to graphs are LLMs-as-Enhancers and LLMs-as-Predictors. LLM-as-Enhancers aim to enhance the quality of node features through the assistance of LLMs. TAPE He et al. (2023) is a groundbreaking example of explanation-based enhancement, encouraging LLMs to produce explanations and pseudo-labels for the augmentation of textual attributes. KEA Chen et al. (2024) instructs LLMs to produce a compilation of knowledge entities, complete with text descriptions, and encodes them using fine-tuned pre-trained language models (PLMs). GLEM Zhao et al. (2022) considers pseudo labels generated by both PLMs and GNNs and incorporates them into a variational EM framework. The fundamental concept of the LLMs-as-Predictors are to leverage LLMs for making predictions in graph-related tasks. InstructGLM Ye et al. (2023) formulates a set of scalable prompts grounded in the maximum hop level and fine-tunes LLMs to output predicted labels directly. GraphLLM Chai et al. (2023) derives the graph-enhanced prefix from the graph representation. This method boosts the LLM's capability in conducting graph reasoning tasks by graph-enhanced prefix tuning. The GraphGPT Tang et al. (2023) framework aligns LLMs with graph structural knowledge using a paradigm of graph instruction tuning.

## B  BENCHMARK DATASETS

To comprehensively and effectively assess the robustness of LLMs in graph learning, we present six text-attributed graphs that offer original textual sentences. For instance, renowned citation graphs such as Cora Sen et al. (2008), PubMed Sen et al. (2008), Citeseer Sen et al. (2008), and ogbn-arxiv (Arxiv) Hu et al. (2020) fall under the category of TAGs. These datasets extract node attributes from textual information, including titles and abstracts of papers. WikiCS Mernyei & Cangea (2020) serves as a Wikipedia-based dataset for benchmarking Graph Neural Networks. It comprises 10 classes corresponding to branches of computer science, demonstrating high connectivity. Node features are derived from the text of the respective articles. Moreover, we employ the History Yan et al. (2023) dataset sourced from Amazon, where node attributes originate from book titles and descriptions. For instance, "Description: Collection of Poetry; Title: The golden treasury of poetry". All datasets are utilized with a low labeling rate split, following the setting described in KEA Chen et al. (2024). The statistics of datasets are reported in Table 9. The licenses of these datasets are MIT License.

Table 9: Statistics of datasets.

|  | Cora | Citeseer | Pubmed | WikiCS | History | ogbn-arxiv |
|---|---|---|---|---|---|---|
| #Nodes | 2,708 | 3,327 | 19,717 | 11,701 | 41,551 | 169,343 |
| #Edges | 5,429 | 4,732 | 44,338 | 216,123 | 358,574 | 1,166,243 |
| #Class | 7 | 6 | 3 | 10 | 12 | 40 |
| #Domain | Academic | Academic | Academic | Wikipedia | E-commerce | Academic |

## C  EXPERIMENTAL SETTINGS AND USAGE INSTRUCTIONS

All algorithms in our benchmark are implemented by PyTorch Paszke et al. (2019). All experiments are conducted on a Linux server with GPU (NVIDIA RTX A6000 48Gb and Tesla V100 32Gb), using PyTorch 2.0.0, PyTorch Geometric 2.4.0 Fey & Lenssen (2019) and Python 3.10.13. We train each model using binary cross-entropy loss, optimized with the Adam optimizer. We use PGD and PRBCD to attack the structures on graphs, which are implemented by DeepRobust Li et al. (2020). For hyperparameter, a hidden size of 256 is used uniformly across all datasets. For the sake of reproducibility, the seeds of random numbers are set to the same. For all the datasets and

models, we tune the following hyper-parameters: learning rate: $lr \in \{0.01, 0.001\}$, weight decay: $\lambda \in \{1e-4, 5e-4\}$.

For further research, our benchmark can easily accommodate the addition of new datasets. Currently, many studies have introduced more graph datasets with textual information. For example, users only need to load the new dataset into the framework in the format of features: x, edges:'edge_index', labels:'y', training set indices:'train_masks', validation set indices:'val_masks', test set indices:'test_masks', and textual attributes: 'raw_texts'. To add new attack and defense methods, users can simply import existing methods from the Deeprobust package. If users wish to use their own attack methods, they can add them to Deeprobust. Finally, our benchmark can advance research on the adversarial robustness of large language models on graphs, allowing users to add new LLM method to the benchmark for use.

For concrete applications, take credit card fraud detection as an example. Fraudsters may manipulate both the network of transactions and the associated textual attributes to evade detection. As researchers build new models for detecting these frauds, our benchmark can be used to quickly evaluate the model's robustness against frauds from both textual and structural levels.

## D    RESULTS OF LLMS-AS-ENHANCERS AGAINST ATTACKS ON GRAPH STRUCTURES

To answer **Q1**, we conduct experiments to examine the robustness against structural attack for LLMs-as-Enhancers. The results are reported in Table 10 and Table 11, which include additional details on the standard deviation (std) compared to Table 1.

Table 10: The performance of LLMs-as-Enhancers against **5%** attacks on graph structures

| | Cora | | | Pudmed | | | Arxiv | | |
|---|---|---|---|---|---|---|---|---|---|
| Feature | Clean | 5% Attack | Gap | Clean | 5% Attack | Gap | Clean | 5% Attack | Gap |
| BOW | 78.49 ± 1.13 | 73.71 ± 1.10 | 03.06% | 74.69 ± 2.07 | 72.40 ± 1.89 | 3.07% | 50.99 ± 2.15 | 42.35 ± 2.64 | 16.94% |
| TFIDF | 81.46 ± 1.21 | 77.23 ± 1.07 | 05.19% | 76.86 ± 1.34 | 74.68 ± 1.40 | 02.84% | 48.39 ± 1.15 | 43.32 ± 1.27 | 10.48% |
| SBert | 81.99 ± 0.76 | 79.18 ± 0.71 | 03.43% | 78.71 ± 1.17 | 74.68 ± 1.40 | 05.12% | 52.51 ± 0.87 | 47.59 ± 1.72 | 09.37% |
| E5 | 83.17 ± 0.73 | 80.43 ± 0.59 | 03.29% | 81.83 ± 1.16 | 79.39 ± 1.06 | 02.98% | 57.04 ± 1.77 | 48.15 ± 1.71 | 15.59% |
| Llama | 78.13 ± 1.07 | 70.91 ± 1.02 | 09.24% | 77.65 ± 0.74 | 76.68 ± 0.94 | 01.25% | 58.04 ± 1.79 | 51.76 ± 1.95 | 10.82% |
| Angle-Llama | 80.28 ± 1.42 | 80.23 ± 0.92 | 0.06% | 75.15 ± 2.50 | 74.65 ± 1.45 | 0.67% | 58.53 ± 1.56 | 51.46 ± 3.33 | 12.08% |
| Explanation | 82.79 ± 1.17 | 80.47 ± 1.28 | 02.80% | 88.84 ± 0.34 | 87.32 ± 0.38 | 01.52% | 54.37 ± 5.51 | 47.19 ± 1.13 | 13.21% |
| Ensemble | 82.57 ± 1.48 | 80.67 ± 1.39 | 02.30% | 83.59 ± 0.81 | 82.55 ± 1.41 | 01.24% | 59.90 ± 2.93 | 54.33 ± 2.43 | 09.30% |
| Llama-FT | 78.54 ± 1.53 | 78.51 ± 0.88 | 0% | 77.40 ± 0.78 | 76.55 ± 0.85 | 1.10% | 52.46 ± 0.84 | 47.59 ± 1.71 | 09.28% |

| | Wikics | | | History | | | Citeseer | | |
|---|---|---|---|---|---|---|---|---|---|
| Feature | Clean | 5% Attack | Gap | Clean | 5% Attack | Gap | Clean | 5% Attack | Gap |
| BOW | 74.92 ± 0.05 | 61.46 ± 0.32 | 17.95% | 51.25 ± 4.80 | 49.04 ± 5.80 | 04.31% | 70.74 ± 0.72 | 68.84 ± 0.82 | 02.69% |
| TFIDF | 75.96 ± 0.14 | 62.03 ± 0.31 | 18.34% | 58.38 ± 3.25 | 57.57 ± 1.69 | 01.39% | 73.02 ± 0.66 | 71.40 ± 0.78 | 02.22% |
| SBert | 75.89 ± 1.77 | 64.44 ± 2.33 | 15.09% | 65.59 ± 1.60 | 65.60 ± 1.48 | 0% | 74.94 ± 0.88 | 72.60 ± 0.81 | 03.14% |
| E5 | 76.71 ± 1.74 | 63.01 ± 2.13 | 17.86% | 66.96 ± 2.30 | 64.69 ± 2.10 | 03.39% | 75.14 ± 0.93 | 72.29 ± 1.27 | 03.79% |
| Llama | 79.72 ± 0.35 | 70.91 ± 1.02 | 11.05% | 66.73 ± 3.53 | 61.64 ± 3.21 | 07.63% | 67.48 ± 3.40 | 70.82 ± 1.07 | 04.95% |
| Angle-Llama | 73.33 ± 1.77 | 67.00 ± 3.62 | 08.63% | 66.52 ± 1.69 | 63.47 ± 3.14 | 04.59% | 71.71 ± 1.73 | 68.46 ± 4.92 | 04.53% |
| Llama-FT | 79.69 ± 0.39 | 70.11 ± 1.03 | 12.02% | 64.53 ± 2.02 | 62.99 ± 1.44 | 02.39% | 69.70 ± 2.18 | 68.62 ± 3.15 | 01.55% |

## E    RESULTS OF LLMS-AS-ENHANCERS AGAINST ATTACKS ON TEXTS

To address **Q2**, we perform a white-box evasion attack on LLMs-as-Enhancers, specifically targeting textual attributes. The results are shown in Table 12, which includes additional details on the standard deviation (std) compared to Table 3.

## F    ANALYSIS FOR TEXTUAL ATTACK

From the experiments on LLMs-as-Enhancers against textual attacks in Section 4.2, we observe that LLaMA performs significantly better than other smaller models, with fine-tuned LLaMA demonstrating the best robustness. Additionally, GCN shows much stronger robustness compared to MLP.

Table 11: The performance of LLMs-as-Enhancers against **25%** attacks on graph structures

| Feature | Cora | | | Pudmed | | | Arxiv | | |
|---|---|---|---|---|---|---|---|---|---|
| | Clean | 25% Attack | Gap | Clean | 25% Attack | Gap | Clean | 25% Attack | Gap |
| BOW | 78.49 ± 1.13 | 60.72 ± 1.92 | 22.64% | 74.69 ± 2.07 | 62.83 ± 1.61 | 15.88% | 50.99 ± 2.15 | 18.95 ± 1.63 | 62.85% |
| TFIDF | 81.46 ± 1.21 | 67.53 ± 0.82 | 17.10% | 76.86 ± 1.34 | 66.18 ± 1.44 | 13.90% | 48.39 ± 1.15 | 24.36 ± 2.10 | 49.67% |
| SBert | 81.99 ± 0.76 | 72.14 ± 2.13 | 12.01% | 78.71 ± 1.17 | 69.24 ± 1.85 | 12.03% | 52.51 ± 0.87 | 29.24 ± 3.27 | 44.32% |
| E5 | 83.17 ± 0.73 | 69.84 ± 0.55 | 16.03% | 81.83 ± 1.16 | 71.06 ± 1.12 | 13.15% | 57.04 ± 1.77 | 25.24 ± 2.77 | 55.75% |
| Llama | 78.13 ± 1.07 | 69.99 ± 1.92 | 10.42% | 77.65 ± 0.72 | 74.62 ± 1.64 | 03.89% | 58.04 ± 1.79 | 31.67 ± 3.09 | 45.43% |
| Angle-Llama | 80.28 ± 1.42 | 76.75 ± 1.33 | 04.40% | 75.15 ± 2.50 | 72.72 ± 3.00 | 03.23% | 58.53 ± 1.56 | 31.73 ± 4.80 | 45.79% |
| Explanation | 82.79 ± 1.17 | 71.32 ± 2.28 | 13.85% | 88.84 ± 0.34 | 82.11 ± 0.72 | 07.57% | 54.37 ± 5.51 | 21.61 ± 3.10 | 60.25% |
| Ensemble | 82.57 ± 1.48 | 74.89 ± 0.93 | 09.30% | 83.59 ± 0.81 | 77.85 ± 0.50 | 06.87% | 59.90 ± 2.93 | 32.82 ± 3.19 | 45.21% |
| Llama-FT | 78.54 ± 1.53 | 70.72 ± 4.15 | 09.96% | 77.40 ± 0.78 | 75.63 ± 2.15 | 2.29% | 52.46 ± 0.84 | 29.19 ± 3.19 | 44.36% |

| Feature | Wikics | | | History | | | Citeseer | | |
|---|---|---|---|---|---|---|---|---|---|
| | Clean | 25% Attack | Gap | Clean | 25% Attack | Gap | Clean | 25% Attack | Gap |
| BOW | 74.92 ± 0.05 | 45.34 ± 0.31 | 39.45% | 51.25 ± 4.80 | 38.93 ± 5.53 | 24.02% | 70.74 ± 0.72 | 61.76 ± 1.44 | 12.69% |
| TFIDF | 75.96 ± 0.14 | 45.74 ± 0.51 | 39.76% | 58.38 ± 3.25 | 22.66 ± 1.69 | 61.15% | 73.02 ± 0.66 | 64.36 ± 0.72 | 11.86% |
| SBert | 75.89 ± 1.77 | 50.19 ± 3.20 | 33.86% | 65.59 ± 1.60 | 56.34 ± 2.41 | 14.10% | 74.94 ± 0.88 | 64.51 ± 1.50 | 13.92% |
| E5 | 76.71 ± 1.74 | 46.38 ± 4.50 | 39.55% | 66.96 ± 2.30 | 55.37 ± 1.80 | 17.30% | 75.14 ± 0.93 | 64.38 ± 1.83 | 14.31% |
| Llama | 79.72 ± 0.35 | 59.23 ± 1.47 | 25.71% | 66.73 ± 3.53 | 54.54 ± 1.74 | 18.26% | 67.48 ± 3.40 | 64.69 ± 2.99 | 04.13% |
| Angle-Llama | 73.33 ± 1.77 | 56.75 ± 4.65 | 22.61% | 66.52 ± 1.69 | 52.75 ± 3.66 | 20.71% | 71.71 ± 1.73 | 68.61 ± 3.61 | 04.32% |
| Llama-FT | 79.69 ± 0.39 | 60.13 ± 0.31 | 24.56% | 64.53 ± 2.02 | 53.34 ± 2.47 | 17.36% | 69.70 ± 2.18 | 66.17 ± 3.62 | 05.07% |

Table 12: Attack success rate (ASR) results of LLMs-as-Enhancers against **5%** textual attack.

| Features | Cora | | PubMed | | Arxiv | |
|---|---|---|---|---|---|---|
| | MLP | GCN | MLP | GCN | MLP | GCN |
| BOW | 62.11±4.48 | 9.27±2.97 | 41.00±7.86 | 8.58±2.45 | 72.69±6.93 | 15.19±4.86 |
| Sbert | 73.18±2.02 | 14.76±2.48 | 45.36±3.88 | 9.32±3.89 | 82.69±10.88 | 11.17±1.39 |
| E5 | 65.29±6.42 | 10.53±1.86 | 35.62±5.64 | 8.76±2.59 | 81.92±8.76 | 15.22±4.18 |
| LLaMA | 56.49±7.34 | 12.58±0.85 | 19.66±4.89 | 4.95±2.59 | 67.87±13.10 | 6.05±2.56 |
| LLaMA-FT | 40.10±12.16 | 4.37±1.94 | 16.69±5.57 | 3.27±1.65 | 67.71±9.83 | 6.08±2.00 |

| Features | Wikics | | History | | Citeseer | |
|---|---|---|---|---|---|---|
| | MLP | GCN | MLP | GCN | MLP | GCN |
| BOW | 67.85±4.88 | 3.82±1.46 | 74.53±10.13 | 18.40±6.00 | 68.80±6.99 | 18.98±4.67 |
| Sbert | 62.12±4.32 | 9.08±2.34 | 76.73±8.79 | 13.41±3.86 | 66.33±6.09 | 16.93±3.31 |
| E5 | 65.99±5.05 | 6.64±2.47 | 61.51±15.43 | 6.54±1.04 | 57.10±6.25 | 14.69±3.73 |
| LLaMA | 22.17±4.96 | 4.17±0.80 | 65.07±9.50 | 15.92±8.47 | 46.81±14.07 | 13.77±5.34 |
| LLaMA-FT | 30.00±2.07 | 2.98±1.83 | 56.98±14.55 | 6.91±3.64 | 38.46±11.19 | 6.58±2.97 |

To explore the reasons behind these observations, we analyze the models from feature and structure perspectives. From the feature perspective, we use SBert and GCN as the victim model and use Latent Dirichlet Allocation to generate the distribution of three themes for each node. Then, we acquire the mean value for each theme. As shown in Fig. 7, the theme distributions for the successfully and failed attacked nodes are similar, indicating that there is not a strong relation between text content and robustness. From the structure perspective, we examine the relationship between the centrality of nodes and the attack success rate. As shown in Fig. 6, we use SBert and GCN as the victim models and visualize the degree distributions of successfully attacked nodes versus failed attacked nodes.

It is evident that successfully attacked nodes often have smaller degrees, indicating that nodes with less structural information in the graph are more vulnerable to textual attacks. Furthermore, similar patterns are observed with eigenvector centrality and PageRank values.

## G  COMBINING TEXTUAL AND STRUCTURAL ATTACK

To enhance attack capabilities, maybe a combined framework that perturbs both text attributes and graph structure is needed. In this study, we provide some preliminary results. First of all, we design a simple strategy that combines the structural attack and textual attack directly, further improving

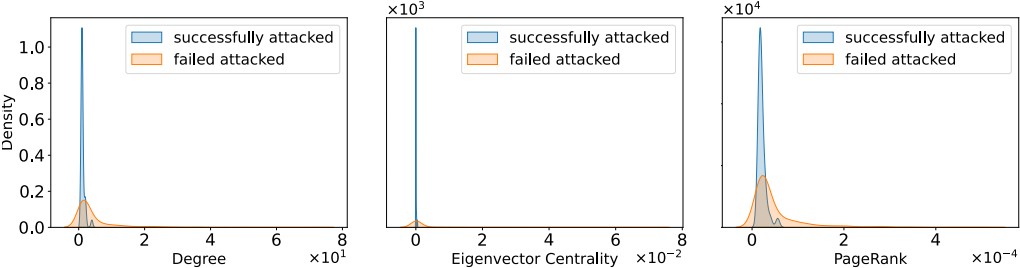

Figure 6: Centrality distributions of the node being attacked successfully and unsuccessfully. We use Sentence-Bert as the victim model and gather all attacked results from the PubMed dataset.

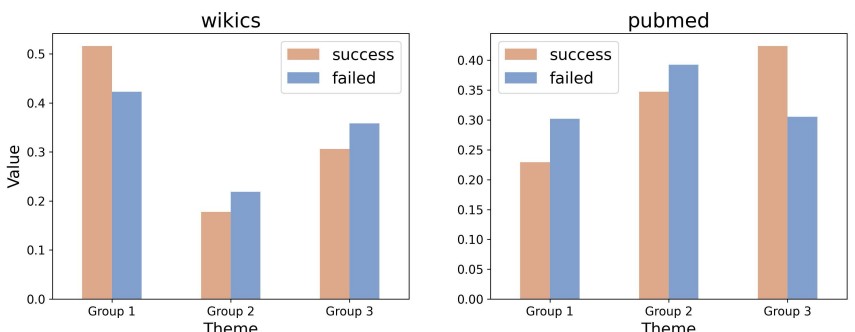

Figure 7: Theme distributions of the node being attacked successfully and unsuccessfully.

the attack ability. The results are reported in Table 13. In addition, we design various combination strategies for modifying the textual attack. These include prioritizing the attack on text attributes of small-degree nodes, targeting large-degree nodes first, and attacking text attributes within the same cluster. Based on the results in Table 14, we find that prioritizing attacks on the text attributes of small-degree nodes is more effective. This is merely a preliminary attempt, and we hope it will inspire more in-depth research.

## H    RESULTS OF GPT-4O MINI AND LLAGA

In addition, we also included results of other predictors: GPT-4o Mini and LLaGA. The results are reported in Table 15 and Table 16.

For textual attack, as the below results show, GPT-4o mini performs robustly against word-level textual-level attacks, aligning with our existing conclusion that LLM exhibits stronger robustness against textual attacks compared to MLP. However, LLaGA only shows competitive performance on PubMed Dataset with MLP.

For structural attack, GPT-4o Mini remains robust compared to GCN and GAT, which aligns with our previous findings on GPT-3.5 and InstructGLM.

## I    ANALYSIS FOR KEY TAKEAWAYS 5

Key Takeaways 5: We found that LLMs-as-Predictors demonstrate strong robustness against structural attacks, particularly in zero-shot settings.

The reason is that all the predictors we examined employed a neighbor sampling strategy, which we believe positively impacts model robustness. To validate this, we applied the same strategy using GraphSAGE against a structural attack (PGD Attack) and observed improved robustness, as shown in Table 17. As the number of adversarial edges increases, the performance of the model without

Table 13: Result of combined attack using SemAttack and PRBCD. Here we perform **5%** perturbation on both text features and the graph structure.

| Feature \ Ptb. | Cora | | Citeseer | | PubMed | | WikiCS | |
|---|---|---|---|---|---|---|---|---|
| | 5% edge | 5% edge & node | 5%edge | 5% edge & node | 5%edge | 5% edge & node | 5%edge | 5% edge & node |
| BOW | 78.40±1.67 | 78.23±1.70 | 69.64±0.98 | 69.58±1.18 | 74.18±0.76 | 74.01±0.78 | 66.38±4.65 | 66.07±4.63 |
| Sbert | 79.93±0.49 | 79.75±0.54 | 74.16±1.23 | 74.15±1.31 | 75.92±1.40 | 75.87±0.90 | 66.94±2.54 | 66.73±2.53 |
| LLaMA | 80.03±0.50 | 79.14±1.17 | 71.70±1.50 | 70.68±1.70 | 75.78±1.33 | 74.95±0.98 | 66.60±5.54 | 66.41±5.64 |

Table 14: Results of **5%** edge and **15%** perturbations under three different sampling strategies on Cora. Small degree first and Large degree first means we sample nodes with the smallest or largest degrees respectively. Clustering refers to sampling nodes that are in the same cluster.

| Cora Dataset | 5% edge & 15% node | |
|---|---|---|
| sampling\perturbation | ASR | ACC |
| Small degree first | 17.20±3.33 | 79.37±0.53 |
| Large degree first | 3.27±1.15 | 79.44±0.85 |
| Clustering | 4.40±1.69 | 79.53±0.61 |

Table 15: The results (ACC) of structure attack (0.05% ptb) on Cora, PubMed, and Citeseer datasets.

| Model | Cora | | | PubMed | | | Citeseer | | |
|---|---|---|---|---|---|---|---|---|---|
| | Clean | Perturbed | GAP | Clean | Perturbed | GAP | Clean | Perturbed | GAP |
| MLP | 69.25 | - | 0% | 78.00 | - | | 68.50 | - | |
| GCN | 81 | 79.75 | 1.54 | 80 | 77.25 | 3.44 | 73.75 | 72.25 | 2.03 |
| GAT | 81.75 | 80.25 | 1.83 | 80.75 | 78 | 3.41 | 74.25 | 73 | 1.68 |
| GPT-4o Mini | 68.56 | 69.31 | -1.14 | 84.56 | 83.63 | 1.07 | 67.04 | 66.38 | 0.79 |

Table 16: The attack success rate (ASR) of word-level textual attack (0.05% ptb) on Cora and PubMed datasets.

| Model | Cora | | | | | PubMed | | | | |
|---|---|---|---|---|---|---|---|---|---|---|
| | full shot | shot=0 | shot=1 | shot=3 | shot=5 | full shot | shot=0 | shot=1 | shot=3 | shot=5 |
| GPT-4o Mini | - | 35.06 | 31.25 | 36.11 | 31.84 | - | 5.72 | 7.71 | 6.76 | 7.52 |
| GPT-3.5 | - | 10.03 | 5.59 | 5.59 | 6.72 | - | 5.53 | 3.75 | 5.29 | 6.07 |
| LLaGA | 8.80 | - | - | - | - | 14.79 | - | - | - | - |
| MLP | 35.97 | - | - | - | - | 14.74 | - | - | - | - |
| GCN | 4.30 | - | - | - | - | 3.20 | - | - | - | - |
| GAT | 2.99 | - | - | - | - | 2.23 | - | - | - | - |

a sampling strategy degrades more rapidly than that of the model with a sampling strategy. The influence of adversarial edges is mitigated because they might not be sampled.

Table 17: Performance (ACC) with and without sampling at various perturbation levels.

| Method | 0 | 0.05 | 0.10 | 0.15 | 0.20 | 0.25 |
|---|---|---|---|---|---|---|
| With Sampling | **81.15 ± 0.99** | **78.85 ± 1.24** | **76.96 ± 1.29** | **76.52 ± 1.45** | **75.87 ± 1.69** | **75.71 ± 1.81** |
| Without Sampling | 81.15 ± 0.99 | 76.86 ± 1.73 | 75.34 ± 1.53 | 74.29 ± 2.07 | 73.91 ± 2.24 | 73.63 ± 1.88 |

## J   RESULTS OF PRBCD AND PGD

From the results in Table 18 and Table 19, we find that the conclusions drawn under PGD and PRBCD attacks for large language models are consistent, with both demonstrating strong robustness.

Table 18: Performance (ACC) for various models on different datasets under PRBCD attack.

| Model | Dataset | GCN/PRBCD | BOW | TFIDF | SBert | E5 | LLaMA |
|---|---|---|---|---|---|---|---|
| Cora | 0 | 77.65 ± 1.34 | 78.01 ± 1.10 | **81.79 ± 0.99** | 81.64 ± 1.23 | 79.96 ± 0.41 | 80.83 ± 0.68 |
| | 5% | 73.22 ± 1.16 | 76.55 ± 0.87 | **79.22 ± 1.58** | 76.66 ± 0.99 | 76.13 ± 1.01 | 76.89 ± 0.82 |
| | 25% | 61.69 ± 1.47 | 67.25 ± 0.67 | **72.31 ± 1.24** | 64.50 ± 2.17 | 62.68 ± 0.62 | 64.27 ± 4.35 |
| Citeseer | 0 | 69.40 ± 0.85 | 69.25 ± 1.09 | **73.68 ± 0.99** | 69.56 ± 0.88 | 68.24 ± 1.30 | 72.53 ± 1.35 |
| | 5% | 63.81 ± 1.39 | 66.85 ± 1.78 | **71.00 ± 1.68** | 69.20 ± 0.73 | 65.88 ± 2.01 | 68.56 ± 1.13 |
| | 25% | 53.32 ± 1.72 | 59.13 ± 1.88 | **63.25 ± 2.32** | 57.74 ± 0.61 | 57.01 ± 2.31 | 56.90 ± 1.02 |
| Pubmed | 0 | 74.89 ± 1.22 | 75.36 ± 1.50 | 77.33 ± 1.43 | **79.35 ± 0.85** | 77.09 ± 1.30 | 74.93 ± 0.95 |
| | 5% | 69.17 ± 1.96 | 72.21 ± 1.25 | 72.39 ± 1.37 | **74.11 ± 1.55** | 70.82 ± 1.90 | 68.47 ± 1.02 |
| | 25% | 54.04 ± 2.04 | 56.53 ± 1.24 | 58.25 ± 2.62 | **60.20 ± 2.32** | 54.09 ± 2.00 | 51.56 ± 2.06 |

Table 19: Performance (ACC) for various models on different datasets under PGD attack.

| Model | Dataset | GCN/PGD | BOW | TFIDF | SBert | E5 | LLaMA |
|---|---|---|---|---|---|---|---|
| Cora | 0 | 78.49 ± 1.13 | 81.46 ± 1.21 | 81.99 ± 0.76 | **83.17 ± 0.73** | 78.13 ± 1.07 | 80.28 ± 1.42 |
| | 5% | 73.71 ± 1.10 | 77.23 ± 1.07 | 79.18 ± 0.71 | **80.43 ± 0.59** | 70.91 ± 1.02 | 80.23 ± 0.92 |
| | 25% | 60.72 ± 1.92 | 67.53 ± 0.82 | 72.14 ± 2.13 | 69.84 ± 0.55 | 69.99 ± 1.92 | **76.75 ± 1.33** |
| Citeseer | 0 | 70.74 ± 0.72 | 73.02 ± 0.66 | 74.94 ± 0.88 | **75.14 ± 0.93** | 67.48 ± 3.40 | 71.71 ± 1.73 |
| | 5% | 68.84 ± 0.82 | 71.40 ± 0.78 | **72.60 ± 0.81** | 72.29 ± 1.27 | 70.82 ± 1.07 | 68.46 ± 4.92 |
| | 25% | 61.76 ± 1.44 | 64.36 ± 0.72 | 64.51 ± 1.50 | 64.38 ± 1.83 | 64.69 ± 2.99 | **68.61 ± 3.61** |
| Pubmed | 0 | 74.69 ± 2.07 | 76.86 ± 1.34 | 78.71 ± 1.17 | **81.83 ± 1.16** | 77.65 ± 0.74 | 75.15 ± 2.50 |
| | 5% | 72.40 ± 1.89 | 74.68 ± 1.40 | 74.68 ± 1.40 | **79.39 ± 1.06** | 76.68 ± 0.94 | 74.65 ± 1.45 |
| | 25% | 62.83 ± 1.61 | 66.18 ± 1.44 | 69.24 ± 1.85 | 71.06 ± 1.12 | **74.62 ± 1.64** | 72.72 ± 3.00 |

## K RESULTS FOR DEEPSEEK

We incorporate additional LLM methods, such as Deepseek, in our experiments. The results are presented in Table 20. As a consistent conclusion, high-quality features generated by models like Deepseek improve model robustness to some extent compared to simple features like BOW and TFIDF.

Table 20: Performance (ACC) for various models on different datasets and perturbation levels by using Deepseek as enhancer.

| Model | Dataset | GCN/PGD | BOW | TFIDF | SBert | E5 | LLaMA | Deepseek |
|---|---|---|---|---|---|---|---|---|
| Cora | 0 | 78.49 | 81.46 | 81.99 | **83.17** | 78.13 | 80.28 | 81.41 |
| | 5% | 73.71 | 77.23 | 79.18 | **80.43** | 70.91 | 80.23 | 78.83 |
| | 25% | 60.72 | 67.53 | 72.14 | 69.84 | 69.99 | **76.75** | 72.48 |
| Citeseer | 0 | 70.74 | 73.02 | 74.94 | **75.14** | 67.48 | 71.71 | 73.12 |
| | 5% | 68.84 | 71.40 | 72.60 | 72.29 | 70.82 | 68.46 | **72.63** |
| | 25% | 61.76 | 64.36 | 64.51 | 64.38 | 64.69 | **68.61** | 65.55 |
| Pubmed | 0 | 74.69 | 76.86 | 78.71 | **81.83** | 77.65 | 75.15 | 76.24 |
| | 5% | 72.40 | 74.68 | 74.68 | **79.39** | 76.68 | 74.65 | 75.27 |
| | 25% | 62.83 | 66.18 | 69.24 | 71.06 | **74.62** | 72.72 | 71.76 |

## L TIME COST

We showcase the computational time required by the LLM to produce high-quality features in Table 21. Simple strategies like BOW and TFIDF mainly utilize indexing to generate features, which is none parametric and have a linear cost. However, E5, Sbert, and Llama all follow transformer architecture, which has a computational cost of $O(N^2)$. Also, for these parametric models, the computational cost increases with the number of parameters. In this paper, the largest model we used

is Llama with 7B parameters and therefore has the highest computational cost. Therefore, using Sbert and E5 offers a more time-efficient solution while still providing strong robustness.

Table 21: Processing time for various models on the Cora dataset.

| Model | Time (s) |
|---|---|
| BOW | 0.4 |
| TFIDF | 0.4 |
| E5 | 57.5 |
| Sbert | 15.5 |
| LLama | 485.6 |
| Angle-LLama | 441.7 |

## M  RESULTS OF ARXIV23

We incorporate additional dataset, such as Arxiv23, into our experiments, where certain portions of this dataset have not been previously encountered by LLMs.

From the results in Table 22, we find that LLaMA demonstrates greater robustness against structural attacks compared to shallow models on Arxiv23.

Table 22: Performance (ACC) for various models on the Arxiv23 dataset at different perturbation levels.

| Dataset | PRBCD | BOW | TFIDF | SBert | E5 | LLaMA |
|---|---|---|---|---|---|---|
| | 0 | 34.12 ± 0.94 | 43.53 ± 2.37 | 55.85 ± 1.49 | 50.88 ± 1.56 | **57.86 ± 1.79** |
| Arxiv23 | 5% | 27.49 ± 1.40 | 38.45 ± 2.76 | 50.69 ± 1.41 | 44.93 ± 1.69 | **55.72 ± 2.54** |
| | 25% | 10.30 ± 0.67 | 18.81 ± 2.41 | 35.38 ± 1.55 | 31.95 ± 1.21 | **40.77 ± 3.74** |

## N  RESULTS FOR SENTENCE-LEVEL ATTACK

For sentence-level attacks, the experimental results are reported in Table 23. These results are consistent with our previous conclusions, which indicate GCN has a stronger robustness against textual attack and LLM-as-Predictor has stronger robustness compared to MLP.

Table 23: The attack success rate (ASR) comparison across datasets under sentence-level attack

| Model | Cora | PubMed | Citeseer |
|---|---|---|---|
| MLP (Sbert embedding) | 46.5 | 32.5 | 60.0 |
| GCN (Sbert embedding) | 4.40 | 4.65 | 6.72 |
| GPT-3.5 zero-shot | 22.78 | 14.00 | 45.79 |
| GPT-3.5 three-shot | 29.69 | 13.38 | 47.96 |
| GPT-4o mini zero-shot | 39.87 | 13.57 | 50.25 |
| GPT-4o mini three-shot | 49.19 | 13.23 | 37.22 |

