# OpenReview forum: "Learning on Graphs with Large Language Models (LLMs): A Deep Dive into Model Robustness"
_ICLR.cc/2025/Conference — Submitted to ICLR 2025_

### Official Review · Reviewer_awX4 · 2024-10-17

**Soundness:** 3
**Presentation:** 2
**Contribution:** 2
**Rating:** 5
**Confidence:** 5

**Summary:**

The paper explores the robustness of Large Language Models (LLMs) when applied to graph adversarial attacks.
It evaluates the ability of LLMs-as-Enhancers and LLMs-as-Predictors to withstand structural and textual perturbations.
Through extensive experiments, the study finds that LLM-based approaches generally outperform traditional shallow models in robustness against both types of attacks.
The research introduces novel benchmarks for evaluating Graph-LLMs’ robustness, provides insights into the underlying factors contributing to LLMs’ superior performance, and offers an open-source benchmark library to promote further research.

**Strengths:**

1. This paper studied an under-explored area, the robustness of Graphs with Large Language Models. The paper presents many interesting insights that are motivating.

2. The experiments are comprehensive, covering both text and structural attacks.

**Weaknesses:**

1. The paper seems more like a benchmark paper but is submitted under the primary area: *learning on graphs and other geometries & topologies*, not *dataset and benchmark*.
If not considered as a benchmark, the novelty of the paper needs further consideration, as the methodology is mostly a combination of previous works.
While still valuable, the results are mostly simple empirical discoveries.

2. It would be better if more fine-tuned models could be included rather than only InstructGLM. I think it is interesting to see more results for the predictors, given that enhancers share mechanisms similar to traditional GNNs.

3. Paper [1] is a very relevant paper that 1) studies textual-level graph adversarial attacks and 2) identifies the robustness of predictors. It would be better to include it for discussion.

4. The presentation could be improved in some areas. For example, the \citep and \citet commands seem to be misused by the authors.


[1] Lei, Runlin, et al. "Intruding with Words: Towards Understanding Graph Injection Attacks at the Text Level." In Neurips 2024.

**Questions:**

1. I'm curious about how to formally define a reasonable attack setting in the era of LLMs.
In the benchmark, multiple settings and budgets are included.
Is it possible to set up a universal attack setting?

2. Under the setting in Question 1, both enhancers and predictors can be utilized by the defender, which facilitates further comparison between them. Is it possible to set up a complete benchmark where both enhancer and predictor can be compared together?
Currently, the benchmark includes too many detailed parts, which might hinder future studies following up.

---

### Official Review · Reviewer_NUXE · 2024-10-28

**Soundness:** 2
**Presentation:** 3
**Contribution:** 2
**Rating:** 5
**Confidence:** 4

**Summary:**

The paper benchmarks the robustness of large language model (LLM)-assisted graph learning. The authors suggest using LLMs both as enhancers and as predictors against structural and textual attacks, finding that LLM-assisted graph learning methods demonstrate better robustness. The topic is interesting and provides insights: the rich knowledge contained in LLMs may help mitigate noise in graph structures and node features, showing promise for developing multimodal graph and language models.

**Strengths:**

Originality: The paper develops a new benchmark for evaluating the robustness of graph-LLM models in adversarial attacks.

Quality: The paper provides comprehensive results for the benchmark with reasonable outcomes.

Clarity: The paper is easy to understand.

Significance: The paper demonstrates a promising future direction in combining LLMs with graph models.

**Weaknesses:**

1. The authors' comprehensive studies and experiments are much appreciated. However, a concern regarding the contributions is that these observations may not be surprising and could be somewhat trivial, as they essentially confirm that LLMs, with their rich knowledge, make graph neural networks (GNNs) more robust to both structural and feature perturbations through improving node features.

2. More actionable observations or conclusions would improve the quality of the paper. For example, in line 538: "This phenomenon may inspire us to design better Graph-LLMs for text attribute encoding." What specific inspiration can we draw from this?

3. It is unclear to what extent (e.g., types of graphs) these observations can be applied, such as to citation graphs, protein-protein interaction graphs, molecular graphs, or social networks.

**Questions:**

1. Providing more context for the structural attacks PGD and PRBCD could improve readability.

2. Line 296: correct ”Explanation”

3. Could the observations/conclusions be applied to broaden the architectural choices of LLMs (e.g., Llama, Qwen, Mistral of various sizes) and graph models (e.g., GCN, GIN, GraphSAGE, GAT)?

---

### Official Review · Reviewer_sPTX · 2024-11-01

**Soundness:** 2
**Presentation:** 2
**Contribution:** 2
**Rating:** 3
**Confidence:** 4

**Summary:**

This paper mainly focuses on evaluating the robustness of GraphLLMs against graph structural and textual perturbation in the term of both LLMs-as-Enhancers and LLMs-as-Predictors.

**Strengths:**

This paper conducts comprehensive evaluation of GraphLLMs robustness under both structural and textual adversarial attacks. They highlight the improved robustness of GraphLLMs as both Enhancers and Predictors. They provide observations and analyses on different factors impacting the model robustness like node degrees.

**Weaknesses:**

1. This paper mainly focuses on robustness evaluation and thus lacks novelty. Developing new attack methods tailored for GraphLLMs or proposing novel defense strategies based on the robustness findings might be interesting.
2. While they evaluate both structural and textual attacks, it lack the full scope of adversarial strategies, like joint structural-textual attack or node injection.
3. The types of evaluation datasets are limited. More datasets can be used like molecular dataset, bioinformatic dataset, social network dataset.
4. This paper only include results of simple architectures like Vanilla GNN and MLP. More complicated models should be evaluated.

**Questions:**

How does the robustness of GraphLLMs vary across different types of graphs, such as social networks and biological networks?

---

### Official Review · Reviewer_dsoE · 2024-11-02

**Soundness:** 2
**Presentation:** 2
**Contribution:** 2
**Rating:** 3
**Confidence:** 3

**Summary:**

The paper investigates the robustness of graph machine learning methods with LLMs (Graph-LLMs) against adversarial attacks. More specifically, the investigation is mainly focused on adversarial attacks of structural and textual perturbations on two kinds of GraphLLM Architectures: LLMs-as-Enhancers and LLMs-as-Predictors. The study reveals some interesting observations about the robustness of Graph-LLMs, and an open-source benchmark library is further released to facilitate further research.

**Strengths:**

1. It seems to be the first work discussing the robustness of Graph-LLMs against adversarial attacks.
2. Extensive experiments are conducted and open-source benchmark library is released to promote reproducibility and enable others to build upon their findings.

**Weaknesses:**

1. As a work aimed to provide a comprehensive benchmark of the robustness of Graph-LLMs against adversarial attacks, the attack evaluated is limited. For both structural attacks and textual attacks, the paper would benefit from a broader range of attack scenarios including: (1) Different knowledge of adversary: white-box, black-box, gray-box, (2) Variety in manipulation techniques: Including adding, deleting, or rewiring of edges, nodes, or texts. For textual attacks, the paper currently only evaluates SemAttack. This leads to the current benchmark cannot provide a thorough understanding of the Graph-LLMs' robustness.
2. The performance metrics include ACC and GAP. It's important to consider which of these better reflects robustness. For example, in Table 1, although LLaMA-FT has a good GAP, the Explanation model achieves the highest ACC under attack, suggesting that this model retains better performance under adversarial conditions, also indicating robustness.
3. I observed an inconsistency in the evaluation metrics used for different types of attacks, with Attack Success Rate being used for textual attacks and GAP for structural attacks. This disparity in metrics may compromise the ability to perform a holistic evaluation of the models' robustness. Adopting a unified metric or a consistent set of metrics across different attack types would facilitate a clearer comparative analysis of results.
4. The paper provides an insightful examination of vulnerabilities in Graph-LLMs; however, it stops short of testing the effectiveness of existing defense strategies against these identified attacks. Including such an analysis would greatly enrich the paper's practical contributions. I recommend that the authors conduct additional experiments to test various defense mechanisms against both textual and structural perturbations.

Minor issues: There is a minor ambiguity in the text, specifically in sentences 201-202 where the term "transferring" is used. It is unclear from the context what exactly is meant to be transferred.

**Questions:**

see Weaknesses

---

### Meta-Review · Area_Chair_HN5Y · 2024-12-20

**Metareview:**

This paper focuses on providing empirical evaluation of LLMs over graphs in terms of its impact on the robustness. The reviewers are concerned about the limited attacks studied in the experiments, the inconsistent metrics used across different experiments, as well as the limited datasets used. Since the main focus of this paper is extensive empirical evaluation, the above comments are important to address. There is no rebuttal provided.

I urge the authors to take the above suggestions into account when preparing the next iteration of the paper.

**Additional Comments On Reviewer Discussion:**

In consensus of rejection.

---

### Decision · Program_Chairs · 2025-01-22

Reject